# Modeling of Disintegration and Dissolution Behavior of Mefenamic Acid Formulation Using Numeric Solution of Noyes-Whitney Equation with Cellular Automata on Microtomographic and Algorithmically Generated Surfaces

**DOI:** 10.3390/pharmaceutics10040259

**Published:** 2018-12-03

**Authors:** Reiji Yokoyama, Go Kimura, Christian M. Schlepütz, Jörg Huwyler, Maxim Puchkov

**Affiliations:** 1Department of Pharmaceutical Sciences, Division of Pharmaceutical Technology, University of Basel, Klingelbergstrasse 50, CH4056 Basel, Switzerland; reiji.yokoyama@shionogi.co.jp (R.Y.); joerg.huwyler@unibas.ch (J.H.); 2Formulation R&D Center, CMC R&D Division, SHIONOGI & CO., LTD., Hyogo 660-0813, Japan; go.kimura@shionogi.co.jp; 3Swiss Light Source, Paul Scherrer Institute, 5232 Villigen, Switzerland; christian.schlepuetz@psi.ch

**Keywords:** drug release simulation, disintegration simulation, poorly water-soluble drug, mefenamic acid, Noyes-Whitney equation, cellular automata, synchrotron microtomography

## Abstract

Manufacturing parameters may have a strong impact on the dissolution and disintegration of solid dosage forms. In line with process analytical technology (PAT) and quality by design approaches, computer-based technologies can be used to design, control, and improve the quality of pharmaceutical compacts and their performance. In view of shortcomings of computationally intensive finite-element or discrete-element methods, we propose a modeling and simulation approach based on numerical solutions of the Noyes-Whitney equation in combination with a cellular automata-supported disintegration model. The results from in vitro release studies of mefenamic acid formulations were compared to calculated release patterns. In silico simulations with our disintegration model showed a high similarity of release profile as compared to the experimental evaluation. Furthermore, algorithmically created virtual tablet structures were in good agreement with microtomography experiments. We conclude that the proposed computational model is a valuable tool to predict the influence of material attributes and process parameters on drug release from tablets.

## 1. Introduction

Many different types of simulation were developed to model the mechanical and dissolution behavior of particles and tablets [1,2] and to also understand the key factors that affect the drug product quality. For example, the finite-element method (FEM), in which the powder is treated as a continuous material, was used to simulate the mechanical behavior, including stress distribution and density distribution, of tablets during compaction [3] and to simulate the drug release from hydrogel-based matrix tablets [4]. The discrete-element method (DEM) was applied to simulate the swelling and dissolution of hydrophilic polymer tablets with different tablet shapes, components, and drug loading [5,6,7,8] and was also applied to characterize the breakage of agglomerates and tablets [9,10]. In addition, DEM and FEM hybrid models were developed for particle breakage and compaction simulations [11,12]. Ideally, DEM requires a large number of virtual particles to take into account the microstructure and component heterogeneity of pharmaceutical tablets [5]. However, it is computationally expensive, due to a requirement to calculate physical and chemical interaction for all individual particles in the system [13]. Therefore, DEM often uses larger spatial subdivisions, i.e., a smaller number of larger particles as compared to those in reality [14]. In addition, knowledge about numerous input parameters is necessary to simulate complex systems with DEM. The DEM model needs to be validated experimentally, which can be difficult [13]. The cellular automata algorithm was proposed as a modeling technique [15,16], and it was applied to simulate the drug release of tablets [17,18]. Three-dimensional cellular algorithms allow the calculation of matrices containing several components organized as a large number of discrete cubes; this is possible due to the simplicity of the calculation, as compared to DEM models [13]. For example, it was reported that the disintegration time of tablets [19], and the buoyancy and drug release profiles of gastroretentive floating tablets [20,21] were simulated with the three-dimensional cellular automata algorithm.

Drug release of immediate-release tablets is influenced by the solubility of the active pharmaceutical ingredient (API) [22,23], particle size distribution [24], granule size, and their arrangement [25,26,27]. In addition, it is well known that the drug release of the pharmaceutical immediate-release tablets can be influenced by the tablet porosity due to the change in disintegration behavior of tablet, i.e., tablet disintegration time increase with a decrease in tablet porosity, resulting in a slow water penetration into tablet [28,29]. Furthermore, in general, the tablet porosity can be variable due to the batch-to-batch difference in compressibility of the powders/granules and variation of compressive stress in the high-speed tablet compaction. Therefore, from the formulation and process development and quality assurance point of view, the development of tools for the computational elucidation of material attributes and the influence of the production process on drug release is very important.

Validation of the simulation results is a challenging topic, mostly arising from an inability to describe the internal structure of a tablet in a sufficiently precise way, i.e., including internal structures at simulation resolution. This challenge is addressed with X-ray computed microtomography, which is a technology to visualize the three-dimensional structure of compacts, and it was applied to visualize the internal pore structure of a tablet and quantitate the density distribution [30,31]. It was also applied to elucidate the correlation between tablet internal structure and dissolution behavior [32,33,34].

The purpose of this study was to simulate the dissolution and disintegration behavior of a poorly soluble drug formulation by applying numeric solutions for dissolution calculation based on the Noyes-Whitney equation and the cellular automata model for tablet disintegration. The influence of the components’ particle arrangement and tablet porosity on drug release profiles was investigated using the simulation of drug release from microtomographic three-dimensional surfaces of real tablets and from arrangements obtained with the three-dimensional cellular automata algorithm. A combined approach for modeling complex multilevel physical processes such as dissolution–disintegration is proposed.

## 2. Materials and Methods

### 2.1. Materials

Mefenamic acid (SIGMA, St. Louis, MO, USA) was used as a model compound to prepare rapidly disintegrating tablets. The excipients used were d-mannitol (Pearlitol 25C, Roquette, Lestrem, France) and microcrystalline cellulose (Avicel PH-101, FMC bioPolymer, Philadelphia, PA, USA) as diluents, croscarmellose sodium (Ac-Di-Sol, FMC bioPolymer, Philadelphia, PA, USA) as a disintegrant, hydroxypropyl cellulose (HPC SL, NIPPON SODA, Tokyo, Japan) as a binder, and magnesium stearate (Peter Greven GmbH & Co, Bad Münstereifel, Germany) as a lubricant. Cetyltrimethylammonium bromide (CTAB) (Merck, Kenilworth, NJ, USA) was used as a surfactant for disintegration tests and dissolution tests.

### 2.2. Methods

#### 2.2.1. Preparation of Tablets

The formulation of the mefenamic acid tablet of 250 mg used for this study was prepared according to Table 1. Mefenamic acid, d-mannitol, microcrystalline cellulose, and croscarmellose sodium were weighed and granulated in a high-shear mixer (MYCROMIX, OYSTAR Hüttlin, Schopfheim, Germany). The powders were pre-mixed for 1 min at an impeller speed of 250 rpm. Next, the granulation process was carried out at 250-rpm impeller speed with a chopper speed of 2000 rpm. Hydroxypropyl cellulose aqueous solution (10% *w*/*w*) was added at a spray rate of approximately 5 g/min. After adding the binding solution, water was added at the same spray rate to flush the line. The process was continued for 1 min. The obtained wet granules were dried and milled using a screen mill (Fitz mill model L1A, Fitz Patrick, Waterloo, ON, Canada). Afterward, the milled granules were mixed with croscarmellose sodium and magnesium stearate as an external phase. The tablets were compressed using a compaction simulator (StylOne, Medel pharma, Beynost, France). The target dwell times for pre-compression and main compression were set to 25 ms. The compaction parameters are shown in Table 1. An 11.28-mm round flat-faced punch set was used for the preparation of the formulation.

#### 2.2.2. Determination of Tablet Porosity and Tensile Strength

Mean tablet weight was evaluated with an electronic balance (AX204 Delta Range, Mettler Toledo, Greifensee, Switzerland). In addition, tablet diameter and tablet thickness were evaluated with a digital caliper (CD-15CPX, Mitutoyo, Kanagawa, Japan). All obtained values of tablet weight, diameter, and thickness were within 1% deviation. True densities of all raw materials were evaluated using helium pycnometry (AccuPyc 1330, Micrometrics, Norcross, GA, USA). The values are given in Table 1.

The true density of tablets was calculated according to Equation (1).
(1)$$\rho_{tablet}=\frac{1}{\sum_{i=1}^{n}\frac{X_{i}}{\rho_{i}}},$$
where *ρ_tablet_* and *ρ_i_* are the true densities (g/cm^3^) of the tablet and each raw material in the tablet, respectively, and *X_i_* is the weight fraction of each component.

The porosity ε of the tablets was determined according to Equation (2).
(2)$$\varepsilon =1-\frac{m}{\pi r^{2}h\rho_{tablet}},$$
where *r* is the tablet radius (mm), and *h* is the tablet thickness (mm).

Tablet hardness was evaluated using a hardness tester (Tablet Tester 8M, Dr. Schleuniger Pharmatron, Allschwil, Switzerland). Tablet hardness can be converted into tensile strength *σ_t_* (MPa), according to Equation (3).
(3)$$\sigma_{t}=\frac{2F}{\pi dh},$$
where *F* is the diametrical crushing force (*N*), and *d* is tablet diameter (mm).

#### 2.2.3. Measurement of Granule Size Distribution

The granule size distribution was measured using a sieve analysis method with a vibrating sieve (Vibro, Retsch, Haan, Germany), equipped with 1000-, 710-, 500-, 355-, 250-, 180-, 125-, and 90-μm sieves.

#### 2.2.4. X-Ray Microtomography

Synchrotron X-ray microtomography measurements of the tablets were performed at the TOMCAT X02DA beamline of the Swiss Light Source at the Paul Scherrer Institute (Villigen, Switzerland). The X-ray beam produced by the superconducting bending magnet source was monochromatized to a beam energy of 19.9 keV using a large bandwidth (ΔE/E~2%) Ru/C multilayer monochromator. Samples were placed in the essentially parallel X-ray beam about 25 m from the source. The radiographic projections of the sample were converted to visible light by a 20-µm-thick LuAG:Ce scintillator coupled to an optical light microscope with a 10-fold magnification (Optique Peter, Lentilly, France), placed 12 mm downstream of the sample to obtain some degree of edge enhancement for phase contrast reconstructions. The magnified image was recorded using a pco.Edge 5.5 sCMOS camera with 2560 × 2160 pixels (h × v) of 6.5 µm in size, resulting in an effective pixel size of 0.65 µm and a field of view (FOV) of 1.66 mm × 1.40 mm (h × v).

The sample tablets’ diameters were significantly larger than the used window for acquisition of projections; therefore, only the central parts of the tablets were reconstructed with the diameter of the section equal to 2.0 mm.

Tomographic reconstructions were computed after applying a single-distance propagation-based phase contrast filter, using a δ/β ratio of 50, with the gridrec reconstruction algorithm employing a standard ramp filter. This resulted in a sufficiently strong contrast between the components’ phases. The volume data were cropped down to 3701 × 3701 pixels in the axial cutting plane during the reconstruction, thus limiting the horizontal extent of the reconstruction to about 2.0 mm.

The preparative processing of the reconstructed data was performed in Image J 1.51j8 (National Institutes of Health, Bethesda, MD, USA) by firstly binning with factor 4 in all dimensions using an averaging function to reduce the memory footprint necessary for effective computation, followed by an SLIC Superpixels [35] clustering analysis for the segmentation of tablet components. In the resulting multipage tagged image file format (TIFF) file, the individual components’ corresponding pixels were mapped according to their types (e.g., for mefenamic acid, the value of 1 was applied) and imported directly into the Particle Arrangement and Compaction module of the software.

#### 2.2.5. Disintegration Test

The disintegration times were measured using a disintegration tester (Sotax DT3, Sotax AG, Allschwil, Switzerland), according to the United States Pharmacopeia (USP) 24 method. Tests were carried out in 900 mL of 50 mM sodium phosphate buffer (pH 6.8) containing 1% CTAB at 37 °C ± 0.5 (*n* = 3). All tests were done in triplicate using six tablets for each test.

#### 2.2.6. Dissolution Test

Dissolution tests of the tablets and the uncompacted granules were carried out using the USP dissolution apparatus II (AT7smart, Sotax, Allschwil, Switzerland) in 900 mL of 50 mM sodium phosphate buffer (pH6.8) containing 1% CTAB at 37 °C ± 0.5 with a paddle rotation of 75 rpm (*n* = 6). Drug concentrations in the dissolution media were measured by an ultraviolet-visible light (UV/Vis) spectrophotometer (Lambda 25, Perkin Elmer, Waltham, MA, USA) at a wavelength of 294 nm every 5 min. The whole amount of uncompacted granules sample was carefully inserted into the dissolution vessel within 5 s of the dissolution test being started.

### 2.3. Simulation of Drug Release with Cellular Automata

#### 2.3.1. Application of Noyes-Whitney Equation in Numeric Calculation of Drug Dissolution

The drug release calculation model was based on the numeric solution of multiparticulate system defined in three-dimensional space according to the tablet geometry. Tablet geometry is approximated by cubic mesh, and the center of mass for all resulting voxels is in the center of every cubic mesh element. To calculate the integral drug release from all voxels, the rate of dissolution *d_m_*/*d_t_* from a voxel element representing a solid drug particle surrounded by solvent voxels under sink conditions can be mathematically described according to Equation (4).
*d_m_*/*d_t_* = (A × D)/λ × (Cs − C), C→0 (sink condition),(4)
where voxel contact surface area A (cm^2^) is $A=\left(1/N\right)\pi \lambda^{2}$, *N* is the number of neighbors (*N* = 26), diffusion coefficient D (cm^2^/s) is according to the Stokes–Einstein relationship [34], Equation (5), Cs is the solubility at equilibrium and at experimental temperature, and C is the concentration of the solid in the bulk of the dissolution medium at time *t*. The diffusion coefficient was calculated according to Equation (5).
(5)$$D=\frac{1}{f}\;\times \;\kappa_{b}\;\times \;\left(T+273.15\right),$$
where frictional coefficient *f* for a sphere given by the Stokes’ law is f=6πηR [36], viscosity of water *η* (Pa·s) is *η* = 2.414 × 10^−5^ × 10^247.8/((T+273.15)−140)^, *k_b_* = 1.3806488 × 10^−16^ (cm^2^·kg·s^−2^·K^−1^) is the Boltzmann constant, *T* (°C) is the temperature, and *R* (Å) is the Stokes radius.

The Stokes–Einstein relation (Equation (5)) is valid only for limited conditions, and was initially applied for ideal gases [37]. To calculate the realistic diffusion of the molecules in the dissolution medium, molecular dynamics simulations were used. We employed Desmond 4.8 (D. E. Shaw Research, New York, NY, USA) molecular dynamics simulations for a single mefenamic acid molecule in the presence of water molecules, at a simulated temperature of 310 K as a standard for dissolution test conditions, using the OPLS_2005 force field, ensemble class NPT (i.e., constant temperature and constant pressure). The system was built with the simple point charge (SPC) water solvent model with neutral charge. System minimization was done with Coulombic interactions with a cut-off of 9.0 Å; the minimization method is a gradient descent with a gradient threshold of 25 kcal/mol/Å and a convergence threshold of 1.0 kcal/mol/Å. The entire root-mean-square deviations (RMSD) of the molecule were recorded, and the diffusion coefficient was calculated from the slope of the sums of squared displacements (Figure 1b) according to Equation (6) [38].
(6)$$D_{s}=\frac{1}{6}\frac{d}{dt}\sum {\langle |r\left(t\right)-r\left(0\right)|\rangle }^{2},$$
where *r*(*t*) is the actual molecular displacement in Å^2^, and *D_S_* is the coefficient of self-diffusion.

The resulting value for the diffusion coefficient of mefenamic acid in water was 3.57 × 10^−7^ cm^2^/s, which is in combination with Equation (4), and the mass of a single drug voxel yields a *C*_1_ value of 22,082 (Table 2), i.e., *C*_1_ is a voxel mass at time 0 divided by the rate of the mass transfer from 1/26th of the voxel surface. This constant is used for convenience during simulation, and is just a simulation software-compatible way of describing dissolution kinetics.

The voxel contact surface area is calculated as 1/26th of the total voxel area, which is calculated as a surface on an inscribed sphere into the grid element. This assumption does not represent the full complexity of an interface contact between liquid a solid; however, it allows for sufficiently accurate mass transfer calculation during simulation. The spatial subdivision of the computed voxel into 26 individual interfaces is according to the Moor-type special neighborhood and is an art of stencil for solving partial differential equations. An increase in the number of used voxels, i.e., a decrease in the voxel size, improves calculation accuracy yet increases computational costs.

Once the rate of mass migration due to dissolution was calculated from Equation (4), the mass of every voxel was calculated during the integration step. The integration was carried out by a simple Euler scheme on 26-point stencil, assuming sufficiently small Δ*t* to minimize error.

To reflect the multicomponent nature of a typical pharmaceutical formulation, which normally consists of several ingredients with different diffusion coefficients and solubilities, the voxels in the dissolution simulation algorithm were assigned specific type information along with physicochemical properties, such as solubility, to calculate mass transfer rates (Equations (4) and (5)). The type information was used while traversing the voxel’s neighborhood and calculating the mass transfer rate for every voxel’s interface, thereby giving an integral difference value for mass migration at a time *t*, reflecting the heterogeneous nature of the calculated system.

The above-described method, despite its apparent simplicity, is very useful for finding the solutions of a mass distribution function, i.e., a distribution of dissolved and undissolved heterogeneous materials at the defined time intervals. However, this approach is only suitable for simulation of non-disintegrating, non-swelling solid pharmaceutical dosage forms, e.g., lozenges [17].

To incorporate an effect of tablet disintegration into the simulation algorithm, the concept of state change was applied to each voxel after calculation of the dissolution rates and integration. The algorithm of disintegration modeling consists of the following four stages:
As soon as a disintegrant cell is signaled to get in contact with a medium-type voxel, its state is converted to “active”.All “active” disintegrant cells mark their direct neighbors for random scattering within the calculation matrix. The labeling depth, i.e., radius around the active disintegrant particles, can be set through the simulations parameter (*C*_2_).All marked cells are randomly distributed in the surrounding medium to maximize the contact surface to the liquid.As soon as the disintegrant cell is “activated”, it loses its action; therefore, the random scattering of its neighborhood can be fired only once.

The dissolution and disintegration algorithms as described above were realized using the modeling software package F-CAD v.2.0, Linux Edition (CINCAP GmbH, Allschwil, Switzerland) applying parallel graphical processing units (Kepler architecture) and dedicated libraries (CUDA 9.1) from Nvidia (Palo Alto, Santa Clara, CA, USA) to reduce the computation time. The simplicity of the proposed algorithm allows its realization on other computational platforms, such as Python or MATLAB, if the usage of the specialized software is restricted. The size of the voxel matrix was set to 330^3^ elements, including solid and dissolution medium voxel types. This calculation matrix size was kept for both types of simulation matrices, that obtained from microtomography and those algorithmically created.

For a comparative analysis between the calculated release pattern obtained from microtomography experiments and the algorithmically created calculation matrix, the latter was constructed by applying cellular automata algorithms for voxel types. To mimic the granular particle arrangement, i.e., to simulate arrangement of the pre-granulated internal phase within tablet constraints, the “Swiss cheese” arrangement procedure was used. This method is described in detail in Reference [17], where the initial granular placeholders are enlarged by sequential application of the CA rule [00101110111111111111111111], where a digit position corresponds to the number of neighbor cells containing the type of interest. In other words, this notation can be transformed into a set of 26 production rules as follows, for example:
Rule: If a cell has three positive neighbors, then, on the next epoch, this cell becomes positive.Rule: If a cell has two positive neighbors, then, on the next epoch, this cell remains unchanged.

The result of the sequential application of the above-stated rule for Moor-type neighborhood on the number of cell triplets randomly distributed in the calculation space results in the development of sphere-like objects. These objects, consisting of multiples of individual cells, start competing for space, thereby eventually forming a non-uniform size distribution. The rule application is stopped as soon as the size distribution reaches the target values, which were set to correspond to the values obtained from real granules after high shear granulation. The acceptance range was set to +/−10% from the average of the real granules.

As soon as the virtual granules were formed, the remaining volumes were blocked by an auxiliary component. The virtual granular material was removed, leaving empty pores in size and shapes of the granules, in other words, negative granules. The voids were filled with randomly distributed cells corresponding to mefenamic acid, d-mannitol, sodium croscarmellose, hydroxypropyl cellulose, and microcrystalline cellulose. The concentrations of the visual material were kept equal to the values from experimentally obtained granules. As the next step, the auxiliary material was removed, and the remaining intergranular voids were filled with the external phase, i.e., remaining amounts of disintegrant and lubricant. In Figure 2, the resulting matrix is shown in contrast to the microCT of the real tablet.

The values used to calculate the dissolution rate constants for mefenamic acid and other formulation components are summarized in the Table 2.

#### 2.3.2. Matrix Arrangement of Tablets

The simulations of the drug release of mefenamic acid tablets were carried out using the software package F-CAD v.2.0. For the simulation of the experimental tablet, the flat-faced round virtual tablets with a diameter of 2 mm were generated. This size was chosen to match the microtomographic acquisition, where the entire tablet scanning was not carried out due to technical limitations. The virtual tablet was discretized into a cubic grid using a voxel side length of 6.5 μm (with 330^3^ elements), equal to the microtomographic resolution with a voxel side length of 6.5 μm.

#### 2.3.3. Comparison of Drug Release Pattern between Experimental and Simulated Profiles

To evaluate the similarity factor (*f*_2_) between simulated and experimental release profiles, Equation (7) was used [39].
(7)$$f_{2}=50\times \log\left\{{\left[1+\frac{1}{n}\sum_{i=1}^{n}{\left(R_{t}-T_{t}\right)}^{2}\right]}^{-0.5}\;\times \;100\right\},$$
where *n* is the number of time points, *R_t_* is the dissolution rate of the experimental tablet at time *t*, and *T_t_* is the dissolution rate of the simulated tablet at time *t*. A similarity factor (*f*_2_) greater than 50 indicates a close correlation between simulated and experimental data.

## 3. Results

### 3.1. In Vitro Evaluation of Drug Release

The properties of the experimental tablets and their compaction condition are summarized in Table 3. In vitro drug release of tablets with different porosities were carried out, and the results are shown in the Figure 3 for uncompacted granules and tablets. To evaluate the differences among the five formulations, the results of one-way ANOVA for formulations A1–A4 and uncompacted granules are given in Table 4. From the results of Table 4, it was suggested that there is a statistically significant difference between release profiles obtained from tablet formulations compacted to different porosities (calculated *F*-values are greater than tabulated *F*-values, and *p* < 0.05).

The obtained similarity factors (*f*_2_) of the drug release among formulations are summarized in Table 5. As can be seen from Figure 3 and Table 5, the similarity factors (*f*_2_) decreased with increased porosity (i.e., formulations A1 to A2 and A3), suggesting that the release rate is influenced by the tablet porosity; however, it is not applicable for all formulations. The uncompressed granules had a very distinctive release profile, quite different from the tablets.

### 3.2. Granule Size Distribution Experimentally Measured and Designed in Simulation Matrices

The granule size distribution of the milled granules is shown in Figure 4.

The granule size distribution was bimodal (the first peak is at 355–500 µm, and the second peak is at 0–90 µm). The sieve fraction of 355–500 µm was dominant relative to any other size fractions; hence, the granule size distributions in the algorithmically created matrices were designed to cover the range of 300–400 µm for formulations A1 to A4.

### 3.3. Comparison between In Vitro and In Silico Drug Release Profiles

The in vitro drug release and in silico drug release of the X-ray CT reconstructed tablets and algorithmically created tablets are shown in Figure 5. The analysis was carried out for A2 and A3 formulations, due to difficulties to distinguish the material of the tablet components for other studied formulations.

The obtained similarity factors (*f*_2_) for the dissolution between the X-ray reconstructed tablet and the experimental tablet were 54 and 72 for formulations A2 and A3, respectively. Also, the obtained similarity factors (*f*_2_) for the dissolution between the algorithmically created matrices and the experimental tablet were 68 and 73 for A2 and A3, respectively. As demonstrated by the similarity factors (*f*_2_), the dissolution profiles from the X-ray reconstructed tablets and algorithmically created matrices were like those obtained from the experimental tablets. It is important to keep in mind that those close correlations are not the results of the fitting, but of ab initio calculations. These results suggest that the simulations of disintegration and dissolution behavior with the calculation matrices obtained from X-ray microtomography are in good agreement with the experimental tablets.

### 3.4. In Silico Evaluation of Drug Release

The in vitro and in silico drug release of algorithmically created matrices are shown in Figure 6 for formulations A1–A4, and the simulated curves describe the experimental data well. Also, the obtained similarity factors (*f*_2_) are summarized in Table 6. The similarity factors (*f*_2_) between in vitro and in silico release were 67, 68, 73, and 71 for formulations A1–A4, respectively, suggesting that the algorithmically created matrices provided a similar drug release to that obtained from the experimental tablets. The simulation dissolution rate calculated with Equation (4) was set to 1.39 × 10^−14^ g/s for a single contact surface of 9.75 × 10^−8^ cm^2^ under an assumption of unstirred layer thickness equal to 2.6 mm. The necessity to use such a large value for an unstirred diffusion layer thickness is dictated by the experimental data [22] and a tendency to produce a cone of powder at the bottom of the dissolution vessel, where mass migration processes are solely diffusion-driven. Unlike the suggested paddle rotation speed [22], the dissolution test was carried out at 75 rpm to reduce the cone effect impact on the release rate [40]. Despite this change, the cone was still clearly observable, and a further increase in the rotation rate would introduce a significant deviation from the literature reference profiles. The similarity of the simulated and the experimental results can be seen for calculation with the disintegration model, whereas the release profiles simulated without the disintegration model resulted in very slow release kinetics (shown in Appendix A). Therefore, this result suggested that the in silico simulation with the disintegration model produced a similar release profile to the in vitro evaluation.

## 4. Discussion

As the in vitro results in Figure 3 show, the fast drug release of the formulation at the beginning of the dissolution test was followed by a slowing down for all studied formulations. This is an expected behavior, which correlates with the known theories for disintegration, where tablet porosity serves as the medium delivery route to cause the disintegrated particles to start swelling [41,42,43]. The release rate from the uncompacted granules is slower than that from the tablets, which is due to the depleted action of the intragranular disintegrant, which was mixed with the other components during wet granulation. The granules were prepared with the wet-granulation method, using a water-based binder solution. In such a case, the internal fraction of the disintegrant was subjected to the liquid contact already during the granulation. The subsequent drying of the produced granules reduced the disintegrating effect of the croscarmellose sodium [44]. By contrast, the tablets also contained croscarmellose sodium in the external phase, which was added to the formulation by powder blending. Therefore, the croscarmellose sodium from the external phase never gets in contact with water, thus retaining its disintegrating ability. Such a difference explains the results obtained experimentally, i.e., the drug release from tablets is faster than from the uncompressed granules. There are literature data indicating the loss of swelling potential of the disintergrants if they are reprocessed by wet granulation [45]. These findings support the choice of the disintegration model simulation’s algorithmic steps. Similar to the existing theories, the “activated” virtual disintegrant cell loses its further potential to force the surrounding components to scatter. In this respect, it is logical to assume that, as soon as all of the disintegrant in the tablets swells, the remaining particles or granules will show different release rates. This effect can be clearly seen in the simulated curves and the experimental results.

As the results in Figure 5 show, the simulated drug release from X-ray reconstructed simulation matrices were like those from the experimental tablets; at the same time, the release from the algorithmically created matrices without disintegration simulation show a less accurate approximation of the experimental data (shown in Appendix A, the maximum drug release below 30% was simulated after 60 min). This result suggests that the disintegration and dissolution models are logically close to real physical behavior and are in good agreement with experiment.

Currently, it is still quite difficult to access the synchrotron X-ray microCT with sufficient contrast phase information to distinguish between the pharmaceutical formulation components. Therefore, in this study, special attention was paid to compare the release profiles between the matrices created algorithmically with the support from cellular automata and from the tablet microtomography. The results in Figure 5 show that the methods to algorithmically construct the calculation matrix can be used to model the experimental drug release if the disintegration model is engaged. The simulated release profiles amplify the bi-phasic nature of the drug release for low-solubility compounds such as mefenamic acid if the formulation contains partially depleted swelling-action disintegrant particles, resulting from water contact during wet high-shear granulation. As soon as the disintegrant in the external phase is used, the release rate drops to the levels observed with the uncompacted granules. This correlates well with the existing disintegration theories [44,45].

As Figure 6 shows, the engagement of the disintegration algorithm in the simulation process results in higher values of similarity factors (*f*_2_) when compared to the simulation without disintegration, including the calculation results from the reconstructed tablets. This suggests that the proposed in silico disintegration procedure is in a good agreement with the experiment. However, the existing deviation between the in vitro and in silico drug release still suggests that there are more subtle mechanisms that contribute to the studied processes, for example, changes in granule particle size during compaction, and percolation effects or wicking between the disintegrant fibers. When comparing the in vitro drug release of formulations A3 and A4 and uncompacted granules as shown in Figure 3, the formulations compacted at higher compressive stresses (i.e., 99 MPa, formulation A3), showed a faster drug release than formulation A4 compacted at lower compressive stress (i.e., 45 MPa). Similar behavior is reported for the uncompressed granules. For this reason, it can be thought that the granules were damaged during the compaction process, and the produced fine fractions resulted in faster drug release profiles. Compressive stress seems to play an important role in maintaining granulometric composition within a tablet, which is well supported in the literature [46,47,48]. However, the damage to granular structures after compressive stress application cannot be seen on the microtomographic acquisitions from these tablets. By contrast, the unchanged granular patterns can be seen in the consecutive cross-sectional images (see horizontal and vertical cross-sectional images shown in Figure 2). Therefore, further investigations to consider the influence of the compression dwelling time, the material mechanical properties, and the behavior of granular breakage may be necessary for a better understanding of the effect of granular partitioning on the disintegration and release rates.

The algorithm for modeling and simulating the disintegration behavior proposed and evaluated in this study can be considered as a tool to elucidate the influences of material attributes of tablets and process parameters on drug release, and can become a useful aid in process and formulation development, especially for bioavailability enhancement of low-solubility compounds, quality assurance, and in general drug product development.

## 5. Conclusions

The disintegration model proposed in this study is a first approximation attempt to construct a comprehensive simulation tool to be used in pharmaceutical development. The proposed model does not feature the fine mechanics of all acting forces being superposed during the wetting, onset on swelling, and final disintegration of the tablet. The used distribution approach of the disintegrated particles is far from reality; however, even these crude approximations improve the simulation model performance.

Future work on accommodating the actions of mechanical forces with either soft particle hydrodynamics or rigid colliding spheres may bring more insight into the fine mechanics of the process of tablet disintegration.

## Figures and Tables

**Figure 1 pharmaceutics-10-00259-f001:**
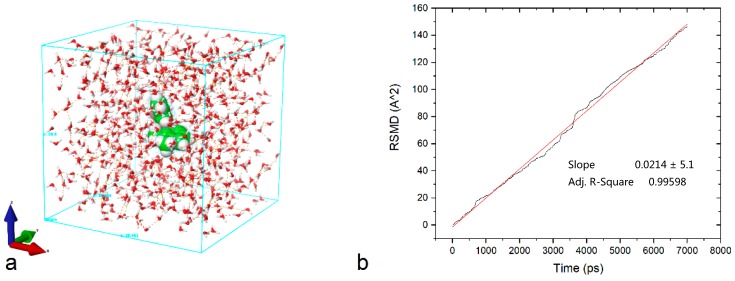
(**a**) The molecular dynamics set-up is shown for simulating the diffusion process of a single molecule of the mefenamic acid in aqueous media; (**b**) the root-mean-squared deviations of the target molecule within 7 ns of simulation time. The slope is the first derivative by time and was used to estimate the diffusion coefficient.

**Figure 2 pharmaceutics-10-00259-f002:**
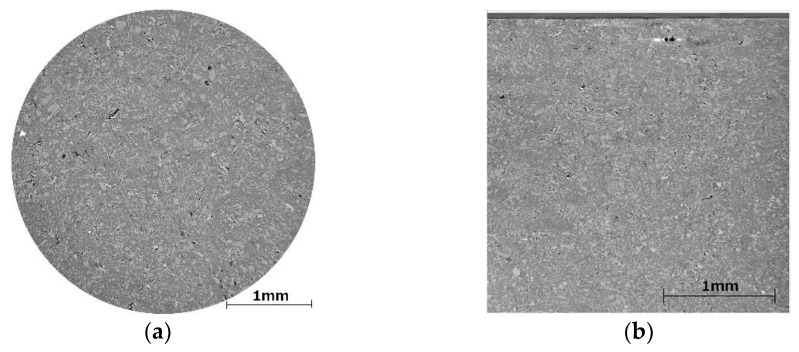
(**a**) The horizontal cross-sectional image of formulation A2 analyzed by microtomography. (**b**) The vertical cross-sectional image of formulation A2 analyzed by microtomography. (**c**) The results of the volume rendering from microtomographic reconstruction for formulation A2, a diameter of 2 mm (red voxel color corresponds to mefenamic acid), and (**d**) algorithmically created component arrangement, a diameter of 2 mm (blue voxels correspond to virtual mefenamic acid particles). (**e**) The skeletonized drawing of the particle distribution (only drug component) is shown after 10 s of simulated dissolution.

**Figure 3 pharmaceutics-10-00259-f003:**
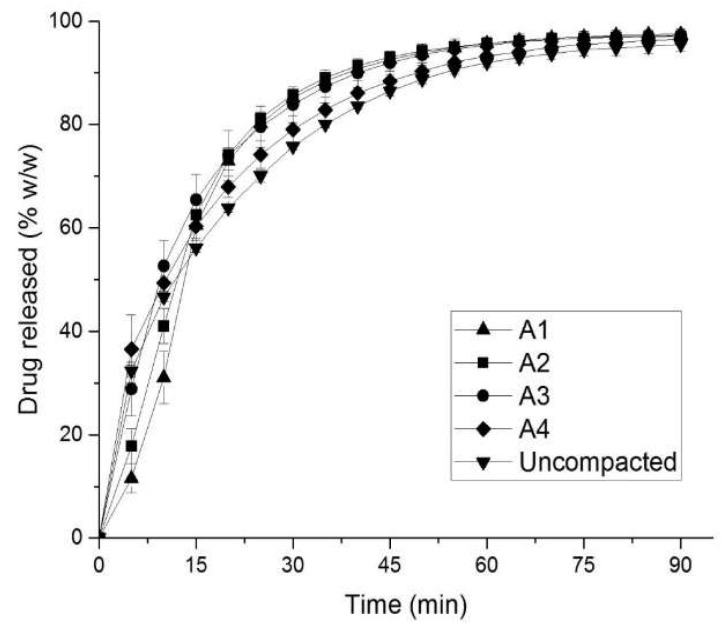
Experimental drug release from mefenamic acid formulations.

**Figure 4 pharmaceutics-10-00259-f004:**
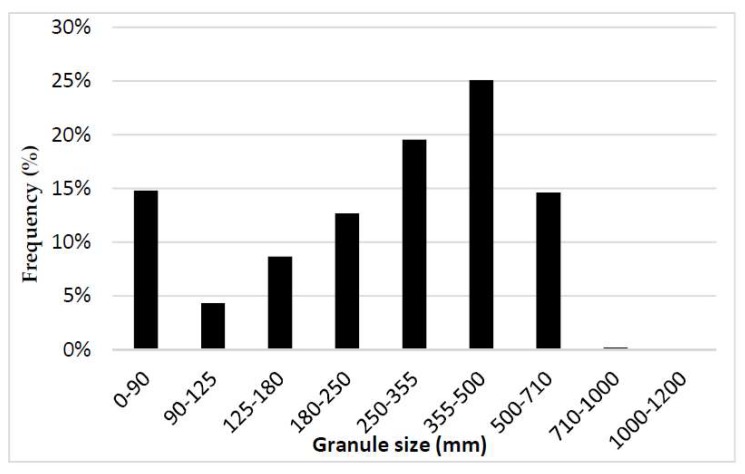
Granule size distribution of mefenamic acid formulation.

**Figure 5 pharmaceutics-10-00259-f005:**
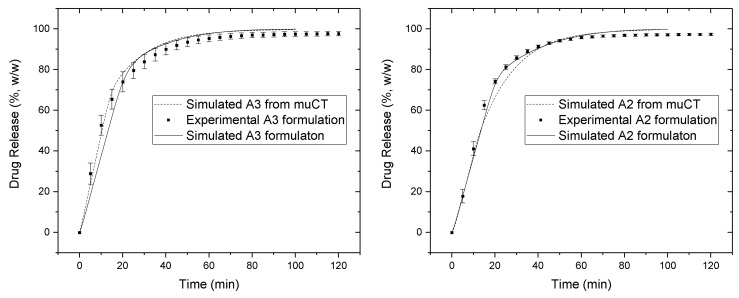
Comparison between in silico and in vitro release profiles obtained from simulations with algorithmically created tablet component arrangements and the reconstructed matrices with the help of microtomography for formulations A3 (**left**) and A2 (**right**).

**Figure 6 pharmaceutics-10-00259-f006:**
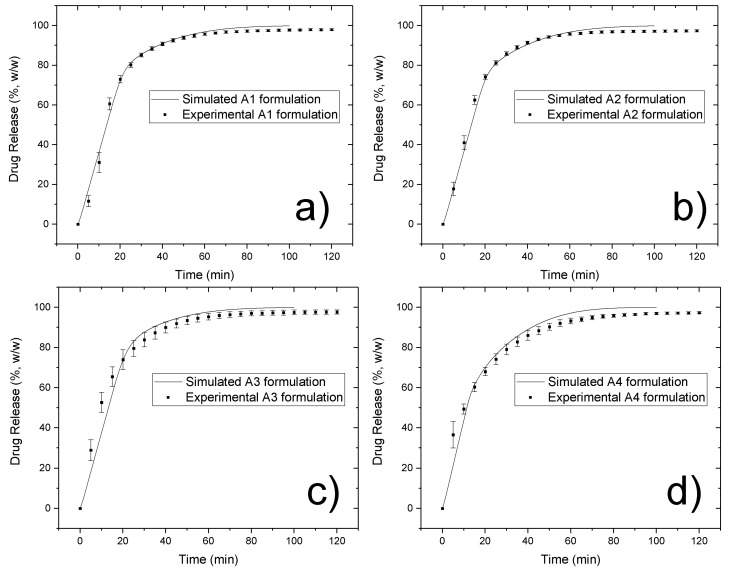
Comparison between simulated release curves obtained from algorithmically created components’ arrangements and corresponding experimental data for formulations A1–A4 (**a**–**d**).

**Table 1 pharmaceutics-10-00259-t001:** Formulation compositions and tablet compaction parameters (A1–A4).

Formulation Composition	True Density (g/cm^3^)	Formulation			
		mg		%, *w*/*w*	
Granular composition					
Mefenamic acid	1.2554	250.0		50.0	
d-mannitol	1.4888	165.0		33.0	
Microcrystalline cellulose	1.5701	50.0		10.0	
Croscarmellose sodium	1.5757	10.0		2.0	
Hydroxypropyl cellulose	1.2334	15.0		3.0	
Granulate	-	490.0		98.0	
External phase composition					
Croscarmellose sodium	1.5757	5.0		1.0	
Magnesium stearate	1.0539	5.0		1.0	
Tablet weight	-	500.0		-	
Tablet Parameters (*N* = 9)	-	A1 *	A2	A3	A4
Tablet porosity (%, *v*/*v*)	-	6	9	14	23
Compressive stress (MPa)	-	210	150	99	45

* Indicates tablet formulation from composition A compressed at 210 MPa.

**Table 2 pharmaceutics-10-00259-t002:** Summary of the parameters used for the in silico dissolution simulation.

Component	True Density (g/cm^3^)	Type Identifier	Component Code	*C*_1_ Constant *	*C*_2_ Constant
Mefenamic acid	1.2554	1	API	22,082	Not used in simulation algorithm for types 1–9
d-Mannitol	1.4888	10	Non swelling, soluble filler	200	Not used in simulation algorithm for types 10–19
Microcrystalline cellulose	1.5701	31	Non-swelling or negligible swelling, insoluble fillers	insoluble	Not used in simulation algorithm for types 30–39
Croscarmellose sodium	1.5757	61	Fibrous disintegrant	insoluble	2 **
Hydroxypropyl cellulose	1.2334	41	Hydrophilic swelling matrix	1 × 10^8^	Swelling of hydrophilic matrix components (types 40–49) was not included into this simulation algorithm
Magnesium stearate	1.0539	71	Hydrophobic ingredient	insoluble	Not used in simulation algorithm for types 70–79

* *C*_1_ reflects the reciprocal dissolution rate of the solid in contact with the simulated dissolution medium (refer to Equations (4)–(6)); ** *C*_2_ indicates the range of disintegration (refer to stage 2 of the disintegration simulation algorithm).

**Table 3 pharmaceutics-10-00259-t003:** Tablet properties and compaction condition.

Formulation	Resultant Compressive Stress (MPa)	Tensile Strength (MPa) (*n* = 3)	Disintegration Time (s) (*n* = 6)	Porosity (%, *v*/*v*)
A1	210	3.31 ± 0.13	543 ± 37	5.6
A2	150	2.53 ± 0.06	311 ± 16	9.5
A3	99	1.48 ± 0.04	160 ± 4	13.7
A4	45	4.72 ± 0.01	53 ± 2	23.1

**Table 4 pharmaceutics-10-00259-t004:** Results of statistical analysis of dissolution rates at 10 min, 15 min, and 30 min for the formulations.

Source of Variance	*F*-Value	*p*-Value	Tabulated *F*-Value
Dissolution rates at 10 min	31.19322	2.18 × 10^−09^ *	2.75871
Dissolution rates at 15 min	7.89681	2.93 × 10^−04^ *	2.75871
Dissolution rates at 30 min	26.35112	1.20 × 10^−08^ *	2.75871

* Statistical significance.

**Table 5 pharmaceutics-10-00259-t005:** Summary of similarity factors (*f*_2_) of the drug release among formulations.

Formulation	A1	A2	A3	A4	Uncompacted Granules
A1	-	61	45	48	48
A2	61	-	57	60	56
A3	45	57	-	67	55
A4	48	60	67	-	73
Uncompacted granules	48	56	55	73	-

**Table 6 pharmaceutics-10-00259-t006:** Summary of similarity factors (*f*_2_) between in vitro and in silico drug release profiles.

Tablet		A1	A2	A3	A4
Similarity factor (*f*_2_)	X-ray reconstructed matrices	NA *	54	72	NA *
	Algorithmically created matrices	67	68	73	71

* Due to difficulties to distinguish the material of the tablet components, the drug release of X-ray reconstructed matrices is not available (NA).

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
