# Peer review of "Modeling of Disintegration and Dissolution Behavior of Mefenamic Acid Formulation Using Numeric Solution of Noyes-Whitney Equation with Cellular Automata on Microtomographic and Algorithmically Generated Surfaces"

_pharmaceutics, 2018, doi:10.3390/pharmaceutics10040259_

Round 1

Reviewer 1 Report

Dear Editor,

this letter provides my feedback as a reviewer of the paper entitled “Investigation of disintegration and dissolution behavior of mefenamic acid drug formulation using numeric solution of Noyes-Whitney equation with cellular automata model on microtomographic surfaces and rational arrangements of tablet components” by Yokoyama and coworkers. With this paper the authors simulate the dissolution and disintegration behaviour of a poorly soluble API applying numeric solutions for dissolution calculation based on Noyes-Whitney equation and cellular automata model for tablet disintegration.

In general, the manuscript is novel and possesses high scientific interest. However, as several part of the paper need clarification I have to recommend for major revision. Hence, I would appreciate if the authors could consider my comments and suggestions, which I think that will be useful for improving the quality of the published work.

- Several typo mistakes need correction (i.e. line 19 “discreet” to “discrete” line 97 “wet-granulates” to “wet-granules”, line 170 “multiarticulate” to “multiparticulate”, line 177 “cm2” the 2 should be in superscript etc.)

- In 2nd paragraph of the introduction the authors should clarify that they are speaking exclusively for immediate release (IR) tablets.

- Line 94-95, please clarify if the HPC solution is in water.

- More details are needed regarding the molecular dynamics simulations employed (force field, charges, energy minimization approach, NVT or NPT etc.). Also, the authors should clarify why they have chosen to perform the MD simulations at 310oC.

- Lines 205-207, please add references proving that the assumption made is indeed valid.

- Line 211: change “with” to “from”

- Lines 215-216: please rephrase the sentence in order to improve readability.

- Lines 220-223: please give references.

- Line 226: The authors should clarify what do they mean by “X stages”.

- Line 299 equation number: Something went wrong with MS-word track-changes. Please, correct.

- Results regarding the true density values, tablet hardness and/or tensile strength are not presented in the manuscript.

- Line 307: Please included the dissolution parameter of uncompressed granules in the material and methods section (how was the granules inserted into the dissolution vessels etc.)

- Line 310: the authors state that: “the release rate is influenced by the tablet porosity”. In order to made such conclusion, the authors should give statistical significance results. Additionally, f2 results among the tested formulations in Figure 3 are needed.

- Line 319: the authors state that: “The granule size distribution is basically single peak”. This is not correct as the particle size distribution is bimodal (a second peak is observed in 0-90 mm) and not unimodal. The authors should elaborate more on this.

- Line 345: There is no Table 4 in the manuscript. (probably a typo for Table 3)

- Lines 350-351: the authors state that: “a tendency to produce a cone of powder at the bottom of the dissolution vessel, where mass migration processes are solely diffusion driven.” How did the author come into such conclusion/assumption? If “cone-effect” is observed during dissolution testing, then the parameter of dissolution process should be adjusted (i.e. increase rotation speed or shift from paddles to baskets). The authors are advised to elaborate more on this matter.

- Line 353-354 and 384: Please, include the not shown data in a supplementary section.

- Table 3: The authors are advised to be more careful during their editing. Porosity results, are given in a table where (based on the caption given) should only include f2 results.

- Line 370: the authors state that “disintegrate particles start to swell”. In the present study croscarmellose sodium was used as disintegrant which based on several textbook water wicking and not swelling is the main mechanism of disintegration. The authors should elaborate on this.

- Lines 370-373: the authors have surprisingly found that the uncompressed granules dissolution rate was slower than the corresponding tables. More elaboration on this is needed.

Author Response

Dear reviewer,

Thank you for the careful review on the manuscript. I comment back to your comments below.

- Several typo mistakes need correction (i.e. line 19 “discreet” to “discrete” line 97 “wet-granulates” to “wet-granules”, line 170 “multiarticulate” to “multiparticulate”, line 177 “cm2” the 2 should be in superscript etc.)

Answer: We are thankful for this careful review of the manuscript. The proposed changes were introduced (Line 101, 175, 182, original line 97, 170 and 177).

- In 2nd paragraph of the introduction the authors should clarify that they are speaking exclusively for immediate release (IR) tablets.

Answer: We agree with this comment and added the word of “immediate release tablets” in 2nd paragraph of the introduction.

- Line 94-95, please clarify if the HPC solution is in water.

Answer: Thank you for your suggestion. We have clarified that HPC aqueous solution was used. (Line 99, original line 94-95)

- More details are needed regarding the molecular dynamics simulations employed (force field, charges, energy minimization approach, NVT or NPT etc.). Also, the authors should clarify why they have chosen to perform the MD simulations at 310oC.

Answer: Methods section the details regarding the molecular dynamic simulation were introduced. (Line 194-199)

- Lines 205-207, please add references proving that the assumption made is indeed valid.

Answer: The following reference was introduced in line 191. (Einstein, A. (1905). "Über die von der molekularkinetischen Theorie der Wärme geforderte Bewegung von in ruhenden Flüssigkeiten suspendierten Teilchen". Annalen der Physik (in German). 322 (8): 549–560)

- Line 211: change “with” to “from”

Answer: Thank you for your suggestion. We have reflected this comment in line 221 (original line 211).

- Lines 215-216: please rephrase the sentence in order to improve readability.

Answer: The sentence was improved, and the corresponding changes were introduced.(Line 224-227, original line 215-216)

- Lines 220-223: please give references.

Answer: The following references were introduced in line 234 to reflect the statement (Puchkov, M. et al, 3-D cellular automata in computer-aided design of pharmaceutical formulations: Mathematical concept and F-CAD software. In Formulation Tools for Pharmaceutical Development; 2013; pp. 155–201 ISBN 9781907568992.)

- Line 226: The authors should clarify what do they mean by “X stages”.

Answer: We are thankful for this comment, the number of stages has been adapted in line 237 (original line 226).

- Line 299 equation number: Something went wrong with MS-word track-changes. Please, correct.

Answer: We are thankful for this careful review of the manuscript. We have corrected the equation number in line 311 (original line 299).

- Results regarding the true density values, tablet hardness and/or tensile strength are not presented in the manuscript.

Answer: We are thankful for this careful review. The results of true density were introduced in Table 1 and the result of tensile strength, disintegration time and porosity were introduced in Table 3.

- Line 307: Please included the dissolution parameter of uncompressed granules in the material and methods section (how was the granules inserted into the dissolution vessels etc.)

Answer: Thank you for providing these insights. We agree that the dissolution test method of uncompacted granules are introduced in the material and methods section (line 166-171).

- Line 310: the authors state that: “the release rate is influenced by the tablet porosity”. In order to made such conclusion, the authors should give statistical significance results. Additionally, f2 results among the tested formulations in Figure 3 are needed.

Answer: Thank you for your suggestion. We have included a new table of the statistical significance results and f2 results in Table 4 and Table 5, respectively.

- Line 319: the authors state that: “The granule size distribution is basically single peak”. This is not correct as the particle size distribution is bimodal (a second peak is observed in 0-90 mm) and not unimodal. The authors should elaborate more on this.

Answer: We agree with you and revised the sentence to explain that the granule size distribution is bimodal. (Line 348-351)

- Line 345: There is no Table 4 in the manuscript. (probably a typo for Table 3)

Answer: We are thankful for this careful review. As commented above, three tables (i.e. Table 3, 4 and 5) were introduced, so the table number was corrected.

- Lines 350-351: the authors state that: “a tendency to produce a cone of powder at the bottom of the dissolution vessel, where mass migration processes are solely diffusion driven.” How did the author come into such conclusion/assumption? If “cone-effect” is observed during dissolution testing, then the parameter of dissolution process should be adjusted (i.e. increase rotation speed or shift from paddles to baskets). The authors are advised to elaborate more on this matter.

Answer: This comment is well taken, the following changes were introduced in line 382-386: “Unlike the suggested paddle rotation speed [23], the dissolution test has been carried out at 75 RPM, to reduce the cone effect impact on the release rate [41]. Despite this change the cone was still well observable and the further increase of the rotation rate would introduce a significant deviation from the literature reference profiles.”

- Line 353-354 and 384: Please, include the not shown data in a supplementary section.

Answer: The supplementary data were added to the manuscript (Line 478).

- Table 3: The authors are advised to be more careful during their editing. Porosity results, are given in a table where (based on the caption given) should only include f2 results.

Answer: Thank you for your suggestion. We removed the porosity results from Table 6 (original Table 3).

- Line 370: the authors state that “disintegrate particles start to swell”. In the present study croscarmellose sodium was used as disintegrant which based on several textbook water wicking and not swelling is the main mechanism of disintegration. The authors should elaborate on this.

Answer: This is an interesting perspective, however, we believe that croscarmellose sodium disintegration mechanism includes both wicking and swelling (although, not as pronounced as, for example, starch based disintegrant) as reported in the literature. In order to refence this behavior, the following reference has been introduced in line 405 (Zhao et al, AAPS Pharm Sci Tech 2005; 6 (1) Article 19.)

- Lines 370-373: the authors have surprisingly found that the uncompressed granules dissolution rate was slower than the corresponding tables. More elaboration on this is needed.

Answer: This is an expected behavior due to method to produce the granules. In order to elaborate this finding the following text has been introduced in line 408-415: “The granules were prepared with the wet-granulation method, using the water-based binder solution. In such case, the internal fraction of the disintegrant has been subjected to the liquid contact already during the granulation. The subsequent drying of the produced granules has reduced the disintegrating effect of the croscarmellose sodium [45]. On contrary, the tablets contain croscarmellose sodium also in the external phase, which is added to the formulation by powder blending. Therefore, the croscarmellose sodium from the external phase never gets in contact with water, thus retains its disintegrating ability intact. Such difference explains the results obtained experimentally, i.e., the drug release from tablets is faster than from the uncompressed granules.”

In addition to the response to the reviewer’s comment, we would like to minor change the title and the abstract in order to make the interests of the manuscript more understandable for the readers.

Best regards,

Reviewer 2 Report

The manuscript by Yokoyama et al describes the disintegration and dissolution models of solid tablets. The results are interesting and could an interest for readers interested in drug design and development. This manuscript can be considered for publication only after the following is addressed.

·      Line 176: change “C_s” to “Cs” consistent with the one in line 178

·      There are two type of D and each has its own formula (equations 5 and 6). Please consider symbol modification to differentiate.

·      There are inconsistencies in writing the scientific notation. Please change “3.57e-7” (line 199), “1e8” (Table 2), “1.39e-14” (line 347) and “9.75e-8” (line 348) to 3.57·10-7, 1·10-8, 1.39·10-14 and 9.75·10-8, respectively, similar to the ones in line 183.

·      In Table 2 (line 285), please explain C1 and C2 constant, and why some C2 constant are not applicable.

·      Table 3 is mentioned in line 305, and yet the table appears in line 362. Please remove the table below the text where it is first mentioned.

·      Please add the description of A1, A2, A3 and A4 from Table 3.

·      The dominant size of granule is between 355 and 500 nm (Figure 4). On what basis, the granule size is determined to be 400 nm (line 321)?

·      Line 370: change “disintegrate particles” to “disintegrated particles”

·      Line 375: change “similarly to” to “similar to”

·      Line 388: change “payed” to “paid”

·      Line 396: add references after “This is well correlated with the existing disintegration theories.”

·      Line 415: change “dwell time” to “dwelling time”

·      Line 428: what is “final dissertation of the tablet”?

·      Line 432: change “insides” to “insights”

·      Be consistent in using either Bristish English or American English throughout the main text, not both. For example, “behavior” in the title is American English but “behaviour” in lines 57, 70, 71, 339, 369, 385, 407, 415, and 418 is British English. “Analyse” in lines 276 and 277 is British English, but “summarize” in lines 284, 305, and 345 is American English. Please check and modify throughout the manuscript.

Author Response

Dear reviewer,

Thank you for your careful review on the manuscript. I comment back to your comment below.

·      Line 176: change “C_s” to “Cs” consistent with the one in line 178

Answer: We are thankful for this careful review. We changed “C_s” to Cs” in line 181(original line 176).

·      There are two type of D and each has its own formula (equations 5 and 6). Please consider symbol modification to differentiate.

Answer: Thank you for your suggestion. The following changes were made in equation 6. (i.e. “D” to “Ds”)

·      There are inconsistencies in writing the scientific notation. Please change “3.57e-7” (line 199), “1e8” (Table 2), “1.39e-14” (line 347) and “9.75e-8” (line 348) to 3.57·10-7, 1·10-8, 1.39·10-14 and 9.75·10-8, respectively, similar to the ones in line 183.

Answer: We are thankful for this careful review. We agree with you and reflected this comment in line 209, 378, 379 and Table 2 (original line 199, 347 and 348, Table2).

·      In Table 2 (line 285), please explain C1 and C2 constant, and why some C2 constant are not applicable.

Answer: This comment is well taken, the corresponding information has been introduced in the Table 2 and the corresponding methods section (lines 227-229 and lines 257-259).

·      Table 3 is mentioned in line 305, and yet the table appears in line 362. Please remove the table below the text where it is first mentioned.

Answer: Thank you for your suggestion. We included a new table of tablet properties in Table 3, and the following table number were corrected.

·      Please add the description of A1, A2, A3 and A4 from Table 3.

Answer: Thank you for your suggestion. We have incorporated your suggestion in lines 397-398 (original 364-365).

·      The dominant size of granule is between 355 and 500 µm (Figure 4). On what basis, the granule size is determined to be 400 nm (line 321)?

Answer: We believe that the granule size distribution with average size between 300 and 400 µm is adequate to simulate the granules which contains the granule size of 355-500 µm as a dominant fraction. The necessary corrections were introduced in line 348-351.

·      Line 370: change “disintegrate particles” to “disintegrated particles”

Answer: We are thankful for this careful review. We reflected this comment.(Line 405)

·      Line 375: change “similarly to” to “similar to”

Answer: We are thankful for this careful review. We reflected this comment.(Line 418)

·      Line 388: change “payed” to “paid”

Answer: We are thankful for this careful review. We reflected this comment. (Line 431)

·      Line 396: add references after “This is well correlated with the existing disintegration theories.”

Answer: Thank you for your suggestion. We added 2 more references indicating the loss of swelling potential of the disintergrants after reprocessing or wet granulation. (Line 439)

·      Line 415: change “dwell time” to “dwelling time”

Answer: We are thankful for this careful review. We reflected this comment. (Line 458)

·      Line 428: what is “final dissertation of the tablet”?

Answer: We are thankful for pinpointing this typing mistake, the manuscript has been updated to “final disintegration of the tablet”. (Line 471)

·      Line 432: change “insides” to “insights”

Answer: We are thankful for this careful review. We reflected this comment. (Line 475)

·      Be consistent in using either Bristish English or American English throughout the main text, not both. For example, “behavior” in the title is American English but “behaviour” in lines 57, 70, 71, 339, 369, 385, 407, 415, and 418 is British English. “Analyse” in lines 276 and 277 is British English, but “summarize” in lines 284, 305, and 345 is American English. Please check and modify throughout the manuscript.

Answer: Thank you for your careful review. We made the following changes to be consistent in using American English. (Title “behavior”, line 177 “Center”, line 287 and 288 “Analyze”)

In addition to the response to the reviewer’s comment, we would like to minor change the title and the abstract in order to make the interests of the manuscript more understandable for the readers.

Best regards,

Round 2

Reviewer 1 Report

Dear Editor,

The present study entitled “Investigation of disintegration and dissolution behavior of mefenamic acid drug formulation using numeric solution of Noyes-Whitney equation with cellular automata model on microtomographic surfaces and rational arrangements of tablet components” by Yokoyama and coworkers describes described the simulation of dissolution and disintegration processes for poorly soluble API based on Noyes-Whitney equation and cellular automata models.

In general, the manuscript is novel and possesses high scientific interest. The authors replied adequately to all comments and suggestions and hence I recommend to be accepted for publication.